# Development and Application of a Human–Machine Interface Using Head Control and Flexible Numeric Tables for the Severely Disabled

**Che-Ming Chang** [1], **Chern-Sheng Lin** [2,*], **Wei-Cheng Chen** [3], **Chung-Ting Chen** [4] **and Yu-Liang Hsu** [2]

[1] Ph.D. Program of Electrical and Communications Engineering, Feng Chia University, Taichung 407, Taiwan; P0800133@o365.fcu.edu.tw

[2] Department of Automatic Control Engineering, Feng Chia University, Taichung 40724, Taiwan; hsuyl@fcu.edu.tw

[3] Master's Program of Biomedical Informatics and Biomedical Engineering, Feng Chia University, Taichung 40724, Taiwan; ja2468102009@hotmail.com

[4] Department of Electronic and Computer Engineering, National Taiwan University of Science and Technology, Taipei 106335, Taiwan; d10802203@mail.ntust.edu.tw

[*] Correspondence: lincs@fcu.edu.tw; Tel.: +886-4-24517250-3929

**Abstract:** The human–machine interface with head control can be applied in many domains. This technology has the valuable application of helping people who cannot use their hands, enabling them to use a computer or speak. This study combines several image processing and computer vision technologies, a digital camera, and software to develop the following system: image processing technologies are adopted to capture the features of head motion; the recognized head gestures include forward, upward, downward, leftward, rightward, right-upper, right-lower, left-upper, and left-lower; corresponding sound modules are used so that patients can communicate with others through a phonetic system and numeric tables. Innovative skin color recognition technology can obtain head features in images. The barycenter of pixels in the feature area is then quickly calculated, and the offset of the barycenter is observed to judge the direction of head motion. This architecture can substantially reduce the distraction of non-targeted objects and enhance the accuracy of systematic judgment.

**Keywords:** computer vision; head gestures; numeric tables

## 1. Introduction

For some people with limb problems, it is impossible to have accurate control over hand movement. Common assistive devices for them include a head-controlled control stick-shaped device, mouth-held stick device, and mouth-controlled blow device. However, these assistive tools are not sanitary, comfortable, or convenient because users have to wear or touch some mechanical sensing devices [1–3]. Given the inconvenience of the existing assistive systems for the disabled, a system, featuring a combination of computer-based vision technology and movement detection, was developed [4–6]. Based on information about the head, it is a household control system for the disabled, and includes color segmentation and head recognition [7].

When the head is held straight without any deflection, the eyes are almost on the same horizontal line; when the head tilts right or left, the angle between the line linking the canthi of the eyes and the horizontal line changes. Therefore, a tilt of the head can be judged according to the angle. There are many algorithms and configurations for head-control devices and sensing human activities [8]. In an

image-type head gesture control device, a CCD (charge-coupled device) camera is used to detect the position of the head without influence from the background. Head movements can also be linked to the position of the cursor on a screen, and the user can control the cursor with his head [9]. Canal developed a system that deals with dynamic gestures, such as waving or nodding, that were recognized using a dynamic time-warping approach based on gesture-specific features computed from depth maps [10]. Lee developed an interface used by severely disabled people who can only move their heads and have difficulties using a joystick, chin stick, and voice- or breath-sensor assistive devices [11]. Terven focused on detecting head-nodding as a relatively simple, non-verbal communication modality because of its significance as a gesture displayed during social interactions [12]. Madeo outlined an approach to automating G-unit and gesture phase segmentation. Their approach treated the segmentation question as a classification problem and used a vector machine with different strategies to solve it [13]. Yi proposed an intelligent wheelchair system based on sEMG (surface electromyography), head gestures, and a feature extraction algorithm based on an improved wavelet packet and sample entropy [14]. Pisharady pointed out the need to consider many measures together with the recognition accuracy of the algorithm to predict its success in real-world applications [15]. Lalithamani used a single-web camera as the input device to recognize hand gestures. Some of these gestures included controlling the mouse cursor, clicking actions, and a few shortcuts for opening specific applications [16]. Halim proposed a method to detect gestures stored in a dictionary with an accuracy of 91% and could define and add custom-made gestures [17].

This study develops a machine vision system. Equipped with a laptop and connected with numeric head-controlled software, it is transformed into a sound-generating system suitable for the disabled. With the system, users can change their head pose, and, using the program code, the change will be converted into acoustic signals to generate sound. This study thus establishes a numeric human–machine interface to provide a more flexible and diverse interface for users.

The rest of this paper is organized as follows: in Section 2, the proposed conceptual design is described in detail. User testing is described in Section 3. Section 4 presents the experimental results and the discussion. Finally, the conclusions are given in Section 5.

## 2. Conceptual Design

This research used image-sensing equipment to detect head rotation position information and complete the comparison of head posture changes. Due to the use of a telescope lens, the user does not need to be very close to the camera to get a clear head image, which can save processing time for face or head detection. A personal computer and a C++ object-oriented development platform were used for program development. In Microsoft Windows, the user's control speed was improved with simple operation methods. In the aspect of searching for the target object (the head), first we adjusted the proper skin color threshold value for image input and, at the same time, calculated the coordinates of the center of the head (Figure 1).

$$\begin{cases} g(x,y) = 1, & (R_{XY} > R_{min} \&\& R_{XY} < R_{max} \&\& G_{XY} > G_{min} \&\& G_{XY} < G_{max} \&\& B_{XY} > B_{min} \&\& B_{XY} < B_{max}) \\ g(x,y) = 0, & otherwise \end{cases} \tag{1}$$

where $R_{XY}$, $G_{XY}$, and $B_{XY}$ are the values of RGB components. *Rmax*, *Rmin*, *Gmax*, *Gmin*, *Bmax*, and *Bmin* are the upper and lower values of RGB channels in the skin image. *g(x,y)* is the new gray level. Equation (1) assumed that in the RGB space skin color is cuboid, which was a simplification.

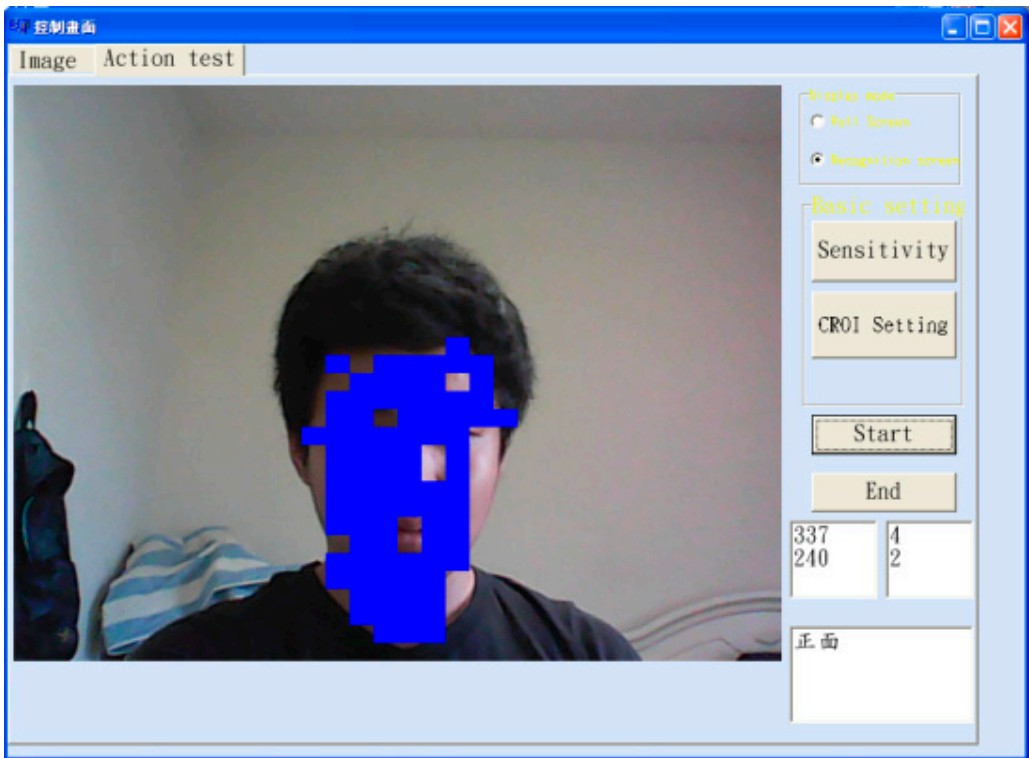

**Figure 1.** The proper skin color threshold result in the image.

We then activated the dynamic image search method and set up a dynamic image search frame by taking the coordinates of the barycenter point of the skin color part in the image (Figure 1) as the initial value. After finding the barycenter point (*Ux, Uy*) in the image, the system proceeded to the validation process.

$$U_x = \frac{\sum_{f=1}^{a} x_f}{a}, \tag{2}$$

$$U_y = \frac{\sum_{f=1}^{a} y_f}{a}, \tag{3}$$

where $x_f$ and $y_f$ are the coordinates of the points of the skin color part in the image. $a$ is the number of pixels in the skin image.

The user had to move along a specific path, according to the coordinates indicated by the system, to confirm that the system had retrieved the feature object. If not, the system must return to the initial setting to retrieve the feature object again. According to the head direction judgment rule of this study, the skin color barycenter point of the front-side image was taken as the datum point [18].

When the head moved, the system recorded the position of skin color points and then judged if the head moved upwards or downwards, according to the change in the position of the skin color points. If the number of skin color points after head movement was smaller than that of the front-side position, the result would be taken as an additional condition for subsequent direction judgment. The barycenter after object movement was then sought in the dynamic image search box. The relevance between the barycenter point after the movement and the new datum point (*Ux₂, Uy₂*) was obtained in the judgment equation, and its corresponding status was displayed (Figure 2) [19].

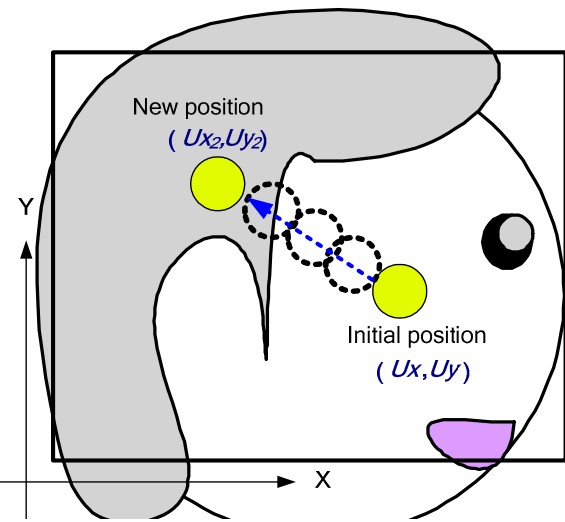

**Figure 2.** The dynamic image search method.

The total number of skin color highlights of the front side of the head was $Ua$, while $Ua_2$ was the total number of skin color highlights after head movement. When $Ua_2$ was smaller than $Ua$, the judgment parameter $Uc$ was set as 0; otherwise it was set as 1.

$$\begin{cases} if\ U_{a2} > U_a\ , U_c = 1 \\ if\ U_{a2} < U_a\ , U_c = 0 \end{cases} \tag{4}$$

Here $(Ux,\ Uy)$ are, respectively, the components $x$ and $y$ of the datum point; $(Ux_2,\ Uy_2)$ are the respective components $x$ and $y$ of the barycenter coordinates of the image after its movement. When the head was raised or lowered, due to light reflection, many pixels belonging to the face were unrecognized as skin, so the system must have had a fault-tolerant design. After the above-mentioned judgment of head direction, the system was able to judge the upward, downward, leftward, and rightward movements of the head. In Figure 3, the blue parts show the facial positions traced by the system. The background was normally the wall of an office room. It can be defined as follows:

$$\begin{cases} if\ U_c = 0\ \&\&\ U_{y2} > U_y + w\ \ \rightarrow down \\ if\ U_c = 0\ \&\&\ U_{y2} < U_y - w\ \ \rightarrow up \\ if\ U_c = 1\ \&\&\ U_{X2} > U_X + w\ \&\&\ U_y - v < U_{y2} < U_y + v\ \ \rightarrow left \\ if\ U_c = 1\ \&\&\ U_{X2} < U_X + w\ \&\&\ U_y - v < U_{y2} < U_y + v\ \ \rightarrow right \end{cases} \tag{5}$$

where $w$, $v$ are the adjustment parameters. It could be extended to judge the right-upward, right-downward, left-upward, and left-downward movements of the head (Figure 4). For initialization, we usually set $w = 20$ and $v = 10$. The skin color area would be much larger when the neck was visible. As the head moved downwards, the area would be smaller. The adjustment parameters could be applied here to solve this problem. The situations in Figure 4 did not show significant differences in blue-area size, but through the innovative judgment method of this system, head movement in eight directions could still be correctly judged.

$$\begin{cases} if\ U_c = 0\ \&\&\ U_x - w < U_{x2} < U_x - v\ \&\&\ U_y - w < U_{y2} < U_y - v\ \rightarrow right\ up \\ if\ U_x = 1\ \&\&\ U_x - w < U_{x2} < U_x - v\ \&\&\ U_y + w > U_{y2} > U_y + v\ \rightarrow right\ down \\ if\ U_c = 1\ \&\&\ U_x + w > U_{x2} > U_x + v\ \&\&\ U_y + w > U_{y2} > U_y + v\ \rightarrow left\ down \\ if\ U_c = 0\ \&\&\ U_x + w > U_{x2} > U_x + v\ \&\&\ U_y - w < U_{y2} < U_y - v\ \rightarrow left\ up \end{cases} \tag{6}$$

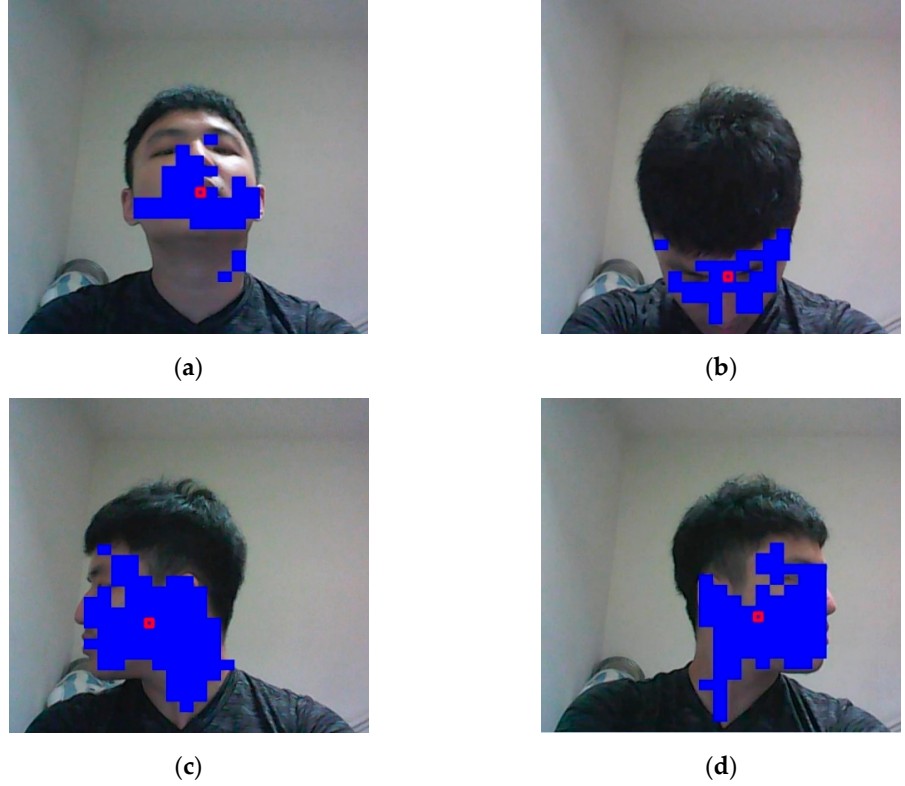

**Figure 3.** Test conditions of the (**a**) upward, (**b**) downward, (**c**) rightward, and (**d**) leftward movements of the head.

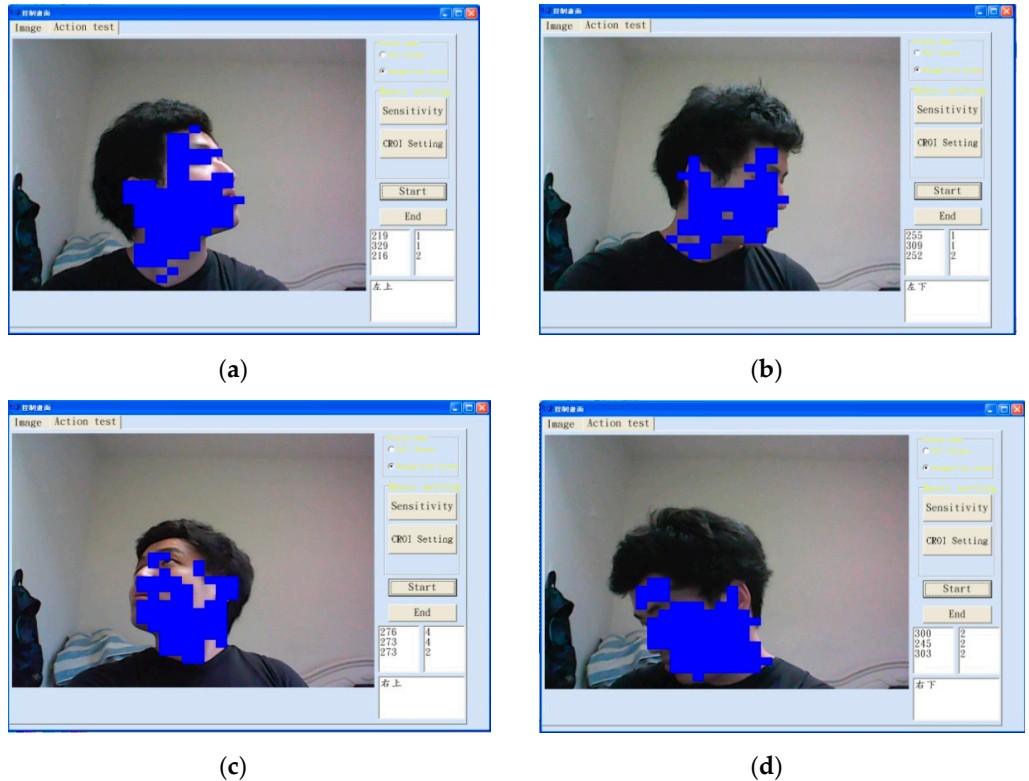

**Figure 4.** Test conditions of the (**a**) left-upward, (**b**) left-downward, (**c**) right-upward, and (**d**) right-downward movements of the head.

## 3. User Testing

Figure 5 shows an example of a Chinese sentence table [20] frequently used in numeric human–machine interfaces. This research established two different Chinese code tables. For the disabled, digitalized Chinese phonetic alphabet tables are far more complicated and difficult to use than the English ones. The selection of the eight-direction model or the four-direction model would lead users to the two-numeric-codes procedure (Figure 6) or the three-numeric-codes procedure (Figure 7). The rules in Chinese reduced the problems that users had with the flexible phonetic alphabet. It could obtain eight postures of up, down, left, right, top right, bottom right, top left, and bottom left, then converted the feature point position result detected by the system for judging. For example, the English letter "E" has a two numeric code of 15. The user needed to complete the first head-action selection "1" and then the second head-action selection "5" to complete the input of the English letter "E". The system did not need an extra activation command, so the user moved their head to select a sentence. When a command was made by mistake, the user needed only to shake his head and try again.

【常用句】

| 51 | 請幫我把腳移到踏板上 | 31 | 請幫我綁安全帶 | 71 | 請你自我介紹，謝謝 |
|---|---|---|---|---|---|
| 52 | 請幫我的腳移開腳踏板 | 32 | 可以這樣做嗎 | 72 | 你有別的想法嗎 |
| 11 | 讓我想一想 | | | 21 | 讓我休息一下 |
| 12 | 請幫我叫媽媽 | | | 22 | 我想喝水，杯子在輪椅後面 |
| 61 | 謝謝你的鼓勵 | 41 | 現在幾點? | 81 | 你決定就好 |
| 62 | 很開心跟你聊天 | 42 | 我可以加你的 Facebook 嗎 | 82 | 這個主意不錯 |

(**a**)

| Sentence Table | | | |
|---|---|---|---|
| 1 | | 2 | |
| 11 | Let me think a minute. | 21 | Can we do it this way? |
| 12 | Please call my mother. | 22 | See you later. |
| 13 | Give me a break. | 23 | What time is it now? |
| 14 | I don't feel well. | 24 | Can we be friend on facebook? |
| 15 | Let me rest a minute | 25 | I have big fish to fry. |
| 16 | I want to take a drink, My cup is in the back of my chair | 26 | Please put my foot on the foot rest. |
| 17 | I'm sorry. | 27 | It sounds like fun. |
| 18 | Please help me fie the seat belt. | 28 | Do me a favor. |
| 3 | | 4 | |
| 31 | Thank you for encouraging me. | 41 | Pardon me! |
| 32 | What's wrong? | 42 | |
| 33 | I appreciate you. | 43 | |
| 34 | Please introduce yourself, Thank you. | 44 | |
| 35 | Do you have another idea? | 45 | |
| 36 | Please talk slowly. | | |
| 37 | Whatever you decide is fine. | | |
| 38 | It's a good suggestion. | | |

(**b**)

**Figure 5.** *Cont.*

| Digital Phonetic Table | | | | | | | | |
|---|---|---|---|---|---|---|---|---|
| **1** | | **2** | | **3** | | **4** | | |
| 11 | A | 21 | H | 31 | O | 41 | T | 71　, |
| 12 | B | 22 | I | 32 | P | 42 | U | 72　. |
| 13 | C | 23 | J | 33 | Q | 43 | V | 77　Chinese |
| 14 | D | 24 | K | 34 | R | 44 | W | 78　Vocabulary |
| 15 | E | 25 | L | 35 | S | 45 | X | 8　Sentence |
| 16 | F | 26 | M | 36　space | | 46 | Y | |
| 17 | G | 27 | N | 37 Block letter | | 47 | Z | |
| 18 | | 28 | | 38　Lower case | | 48 abbreviation | | |

(In the bottom-right cell: a grid reading 5, 3, 7 / 1, (face), 2 / 6, 4, 8)

**(c)**

**Figure 5.** The flexible designs of frequently used (**a**) Chinese and (**b**) English sentences, and (**c**) English digital phonetic table of a numeric human–machine interface.

| Consonant | | | | | | | |
|---|---|---|---|---|---|---|---|
| ㄅ : 12 | ㄆ : 13 | ㄇ : 14 | ㄈ : 21 | ㄉ : 22 | ㄊ : 23 | ㄋ : 24 | ㄌ : 31 |
| ㄍ : 32 | ㄎ : 33 | ㄏ : 34 | ㄐ : 41 | ㄑ : 42 | ㄒ : 43 | ㄓ : 44 | ㄔ : 51 |
| ㄕ : 52 | ㄖ : 53 | ㄗ : 54 | ㄘ : 61 | ㄙ : 62 | none : 11 | | |

| Intermediate sign | | | |
|---|---|---|---|
| ㄧ : 12 | ㄨ : 13 | ㄩ : 14 | nonc : 11 |

| Vowel | | | | | | |
|---|---|---|---|---|---|---|
| ㄚ : 12 | ㄛ : 13 | ㄜ : 14 | ㄝ : 21 | ㄞ : 22 | ㄟ : 23 | ㄠ : 24 |
| ㄡ : 31 | ㄢ : 32 | ㄣ : 33 | ㄤ : 34 | ㄥ : 41 | ㄦ : 42 | none: 11 |

| Tone | | | | |
|---|---|---|---|---|
| none : 12 | ˙ : 13 | ´ : 14 | ˇ : 21 | ` : 22 |

**Figure 6.** Two-numeric-code table of the Chinese phonetic alphabets.

| Consonant | | | | | | | |
|---|---|---|---|---|---|---|---|
| ㄅ :112 | ㄆ :113 | ㄇ :114 | ㄈ :121 | ㄉ :122 | ㄊ :123 | ㄋ :124 | ㄌ :131 |
| ㄍ :132 | ㄎ :133 | ㄏ :134 | ㄐ :141 | ㄑ :142 | ㄒ :143 | ㄓ :144 | ㄔ :211 |
| ㄕ :212 | ㄖ :213 | ㄗ :214 | ㄘ :221 | ㄙ :222 | none : 111 | | |

| Intermediate sign | | | |
|---|---|---|---|
| ㄧ : 112 | ㄨ : 113 | ㄩ : 114 | none : 111 |

| Vowel | | | | | | |
|---|---|---|---|---|---|---|
| ㄚ : 112 | ㄛ : 113 | ㄜ : 114 | ㄝ : 121 | ㄞ : 122 | ㄟ : 123 | ㄠ : 124 |
| ㄡ : 131 | ㄢ : 132 | ㄣ : 133 | ㄤ : 134 | ㄥ : 141 | ㄦ : 142 | none : 111 |

| Tone | | | | |
|---|---|---|---|---|
| none : 112 | • : 113 | ´ : 114 | ˇ : 121 | ` : 122 |

**Figure 7.** Three-numeric-code table of the Chinese phonetic alphabet.

The test results of computing position stability of this study are shown in Figure 8. For the analysis of the barycenter algorithm, the experiment placed the black center point in the center of the image. The actual position coordinates in the image captured by the camera were compared with barycenter

position coordinates. The experiment was repeated 50 times, and, according to Figure 8, the accuracy of the barycenter algorithm was high; hence, the human–machine interface of the system resulted in a low probability of misjudgment and was convenient to operate.

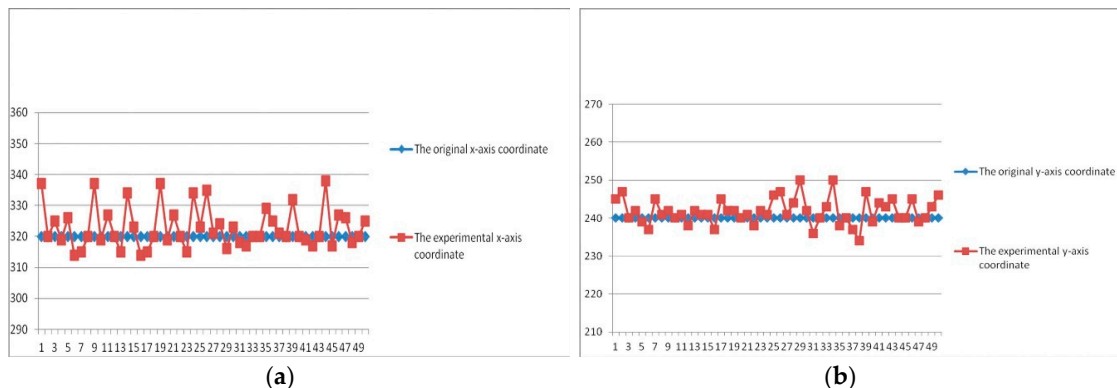

**Figure 8.** The test results of computing position stability of this study; (**a**) *x* axis coordinate; (**b**) *y* axis coordinate.

This study included statistical analysis of the accuracy of the head-pose-detection system. Through the conversion of the eight poses (up, down, left, right, upper right, lower right, upper left, and lower left), the feature point positions detected by the system and the judgment-based positions were observed. The eight-direction and four-direction systems were both tested 80 times under three states: (1) finish one movement within 10 s; (2) finish one movement within 5 s; and (3) finish one movement within 2 s. The results are shown in Figures 9 and 10.

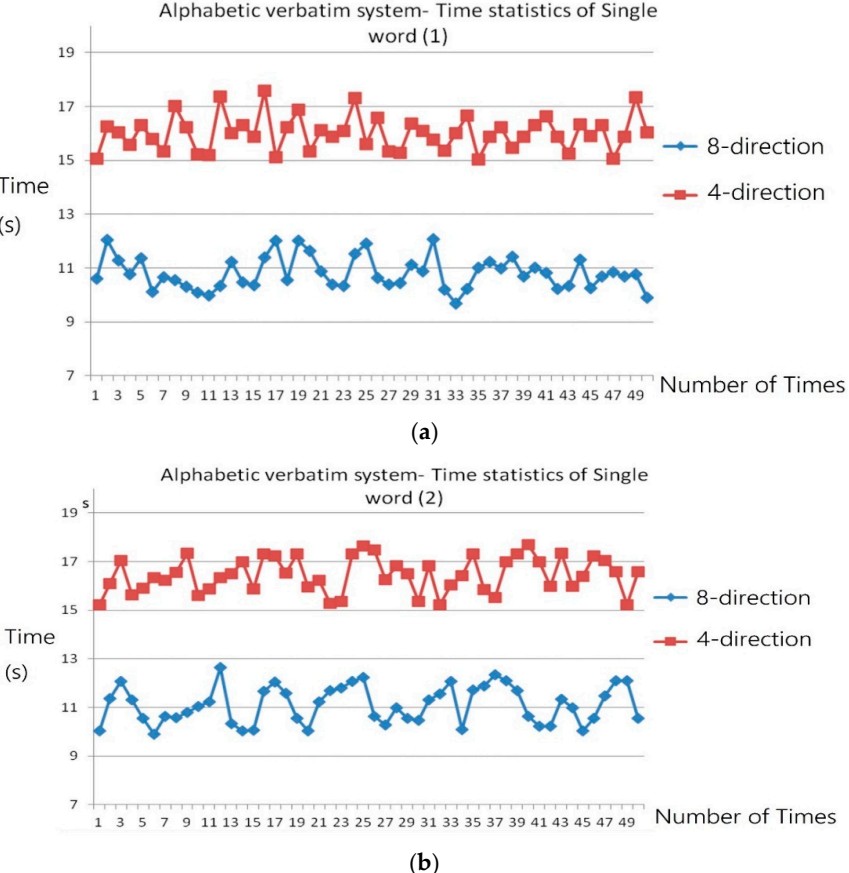

**Figure 9.** *Cont.*

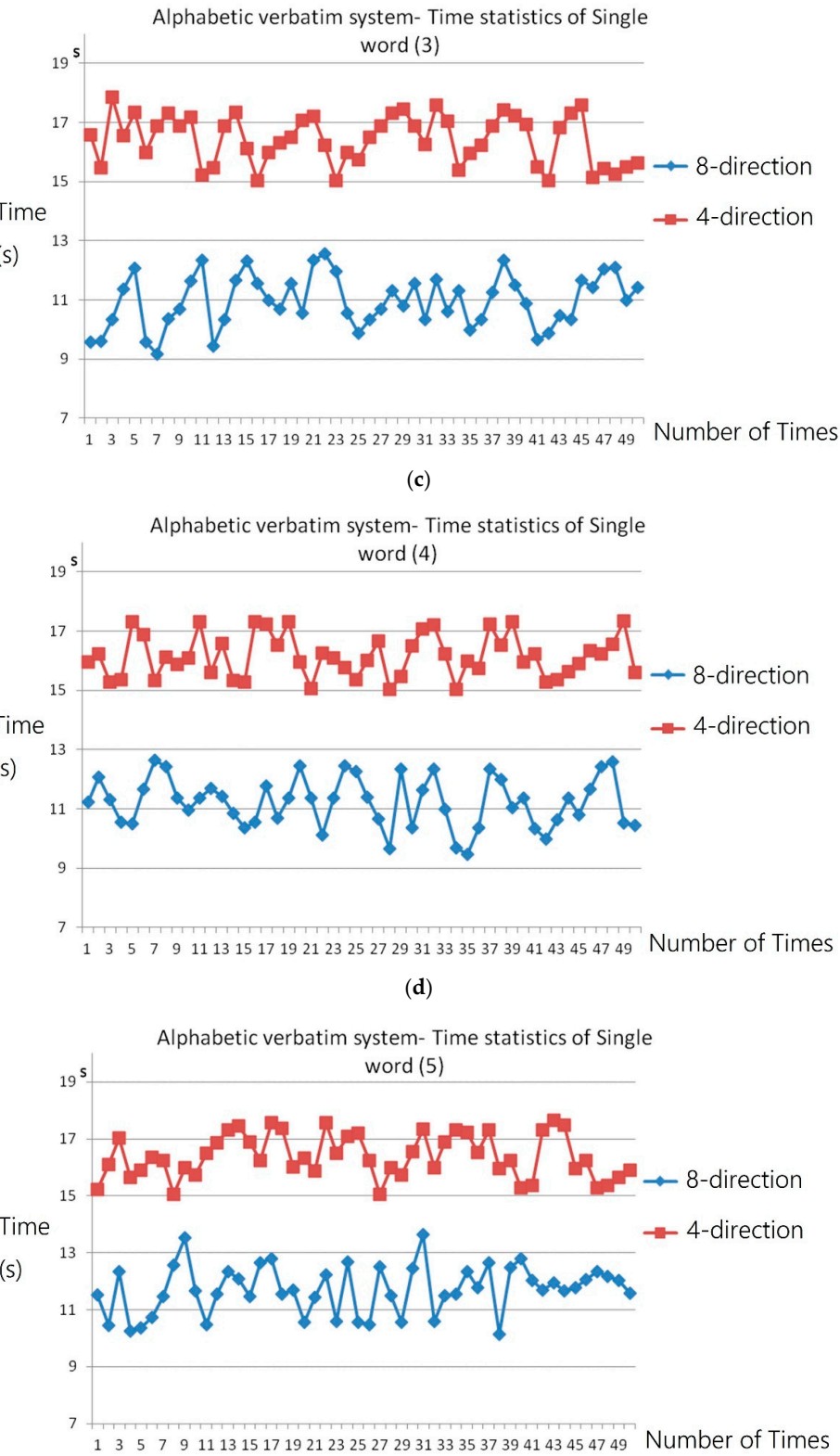

**Figure 9.** *Cont.*

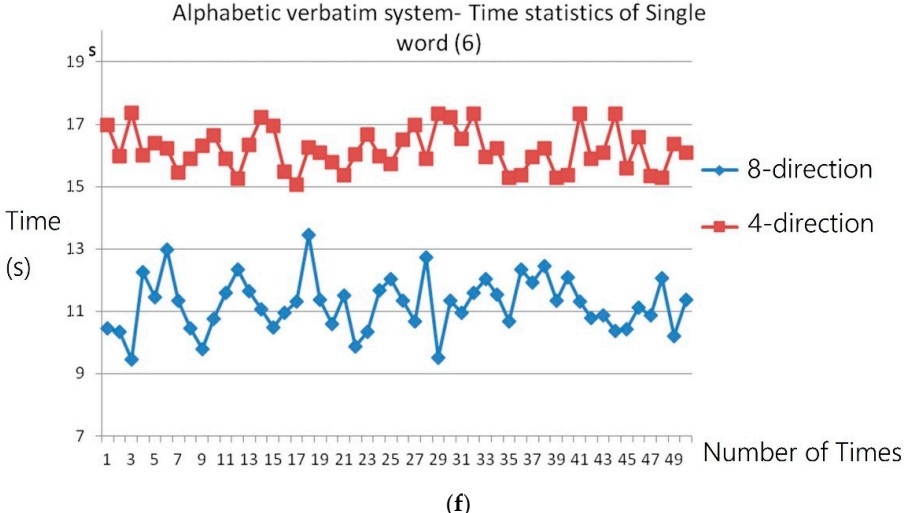

(**f**)

**Figure 9.** The test results of the numeric Chinese phonetic alphabet table with head-control interface (**a**) user #1 (**b**) user #2 (**c**) user #3 (**d**) user #4 (**e**) user #5 (**f**) user #6.

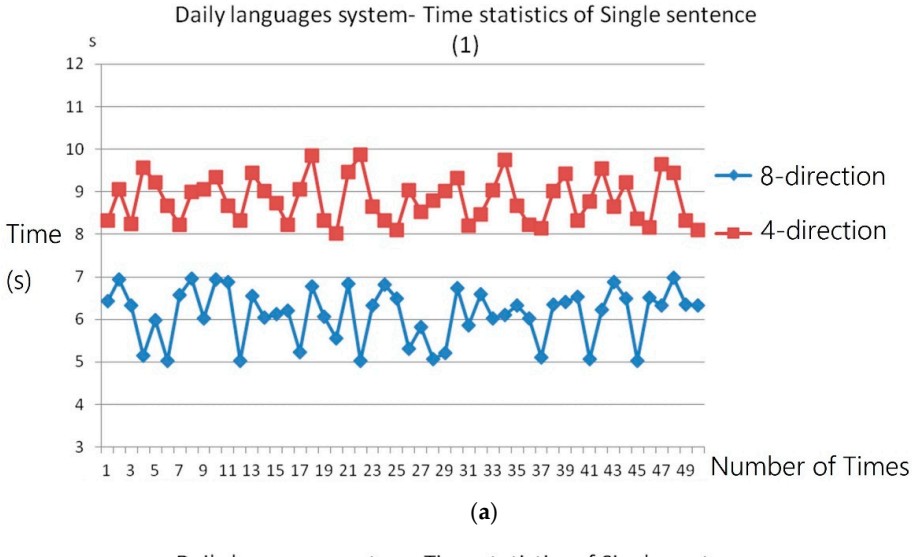

(**a**)

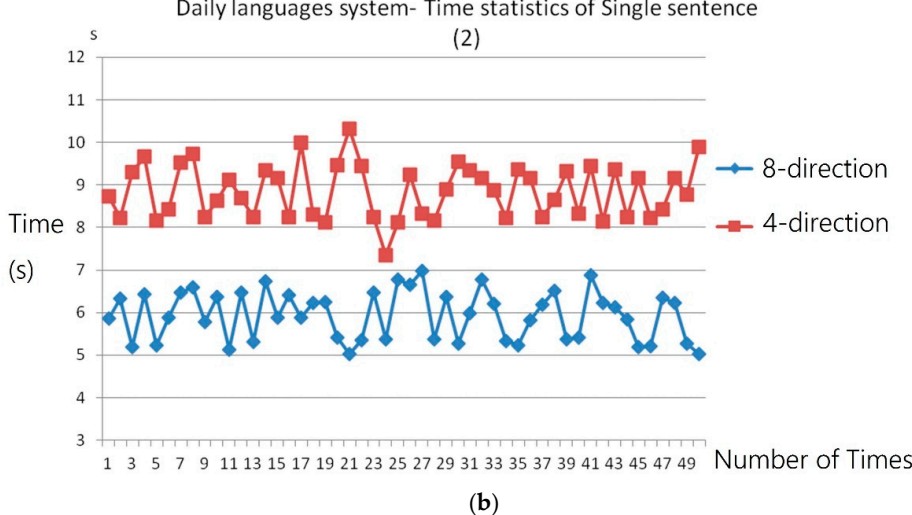

(**b**)

**Figure 10.** *Cont.*

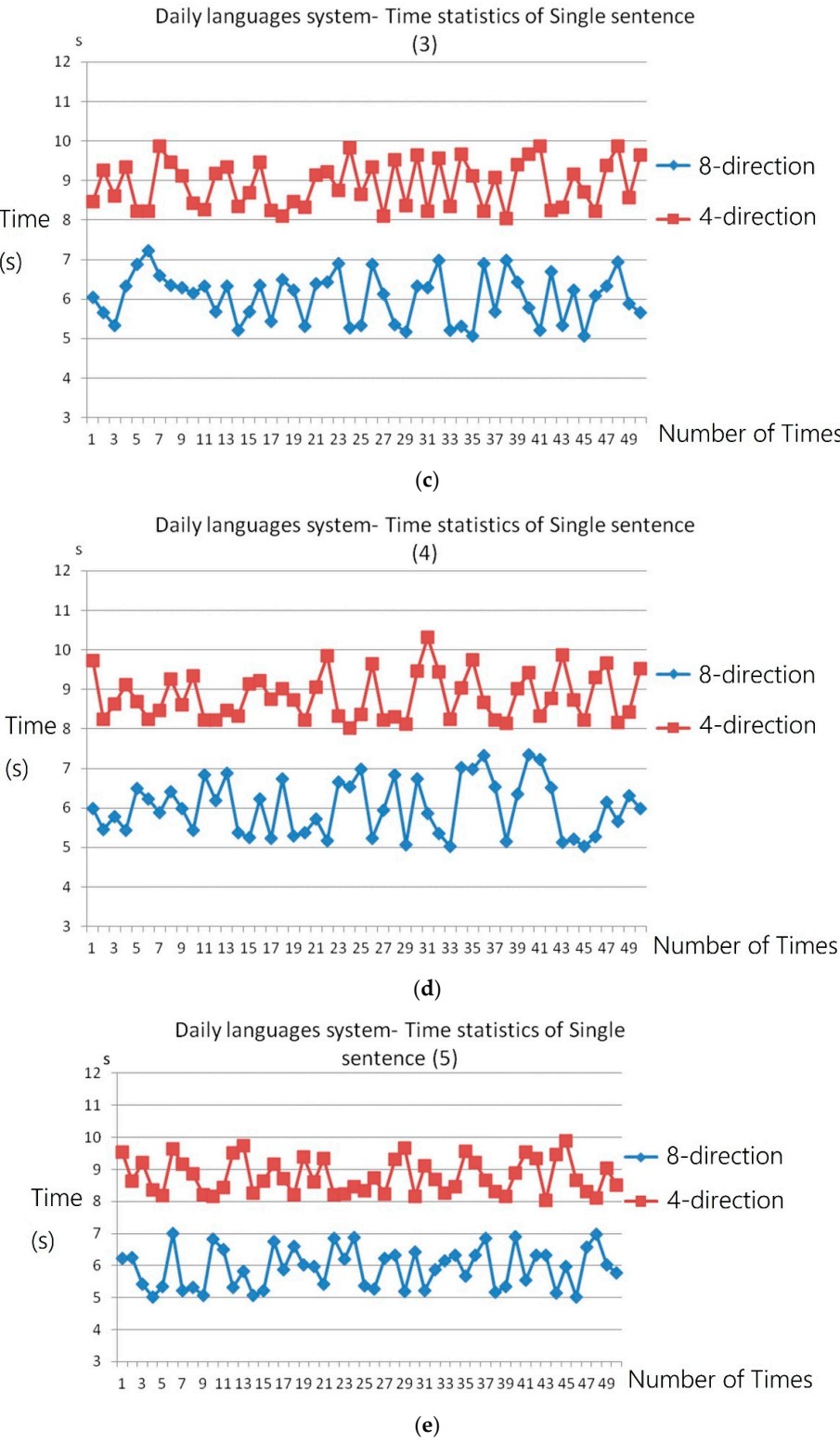

**Figure 10.** *Cont.*

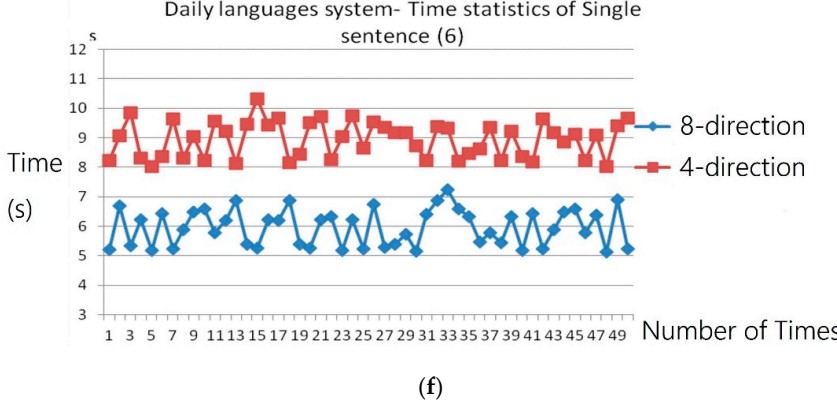

(**f**)

**Figure 10.** The test results of the frequently used Chinese numeric sentences table with head-control interface (**a**) user #1 (**b**) user #2 (**c**) user #3 (**d**) user #4 (**e**) user #5 (**f**) user #6.

The scope of the direction judgment in the experiment was as follows: up, $y < 220$; down, $y > 370$; left, $x > 375$; right, $x < 310$; lower right, $x < 310$ and $y > 370$; upper right, $x < 310$ and $y < 220$; lower left, $x > 375$ and $y > 370$; upper left, $x < 310$ and $y < 220$; x and y are the respective horizontal and vertical coordinate points of the barycenter. According to the statistical results, the accuracy of one movement in the eight directions within 10 s was 100%, 97.5% within 5 s, and 92.5% within 2 s. The accuracy of one movement in the four directions was 100% within 10 s and 5s, and 97.5% within 2 s. This demonstrated that if users required to finish a head turn within 2 s with the system's help, there might be errors due to inadequate time for procedural judgment. Nonetheless, the accuracy was still over 90%. Therefore, the system was quite stable.

The experiment of coding time count consisted of two parts:

1. The average time for each word in the word-by-word sound-generating system;
2. The time for a sentence in the frequently used sentence system.

The two parts included the four-direction and eight-direction codes, respectively. Six users and their family members were invited to participate in the experiment. In the word-by-word sound-generating system, the words with three different syllables were taken as the testing words, and "Please help me." was taken as the testing sentence for the pronunciation of frequently used sentences [20–22]. The experiment results showed the average time the system required to spell a word and to finish a frequently used sentence. Recording of the time started with the user's head turn and did not stop until the system finished the coding and generated sound.

## 4. Experimental Results and Discussion

According to the experiment results, the average time for the word-by-word sound-generating system to spell a word was as follows: (1) eight directions = 11.17 ± 0.85 s; (2) four directions = 16.27 ± 0.74 s. The average time for the frequently used sentence system to finish a sentence was as follows: eight directions = 6.00 ± 0.63 s; (3) four directions = 8.86 ± 0.58 s.

With the users' experience and their suggestions, this study improved the head-controlled system. Aside from testing the system, the patients also participated in the experiment of coding time count. With head-turn images filmed by the CCD and image scanning, this study obtained the parameters of a head turn and facilitated position detection. The obtained parameters were used as coding elements in a phonetic system. As head feature colors could be captured, it was possible to locate the head without the parameters set in the procedure. This not only simplified parameter adjustment, but also enabled the camera to locate the head at any time.

In this study, different test environments and scenes were taken, and test results were used as adjustment coefficients in image processing. After the tests, the image processing and environment control methods most suitable for the system were found. The methods could lead to the correct

judgment of head-turn directions. The system was used for repeated experiments, and the misjudgment rate was less than 3%, which meant that the system was highly stable.

For the disabled who cannot accurately operate the bi-system coding of four or eight directions, there was a probability of misjudgment, caused by their inability to control their bodies. In the future, it is suggested that image judgment should be upgraded to make the program more complete. Additionally, the existing four-direction coding is too difficult for Chinese grammatical coding. If it could be extended to the eight-direction judgment for coding, then the combination of the recognized data and the phonetic system would allow users to operate the system more conveniently.

The detection and coding of the system could also be applied to a rehabilitation system for the disabled. Unlike the traditional method, where the architecture is specifically designed for different movement control training, the combination of computer vision technology and movement control training enables users to train themselves with head turn games, makes rehabilitation training more attractive, and equips rehabilitation with both training and entertainment.

## 5. Conclusions

This study developed a system where machine vision technology was integrated with action-control for display on a numeric human–machine interface and applied it to a phonetic system for patients with general paralysis. The system used machine vision technology to detect eight directions of head motion and displayed the obtained direction data on the human–machine interface. This was then combined with a phonetic module to develop software that generated sound according to head control. Differently from traditional supporting devices for those with physical problems, the interactive head-control device featuring the combination of machine vision technology and action recognition was both practical and convenient. This device had many applications in complex and economical equipment, as confirmed by its use in environmental control. We hope our head-control device can become more useful for human–machine interfaces in the future.

**Author Contributions:** Conceptualization, C.-M.C., C.-S.L., W.-C.C., C.-T.C. and Y.-L.H.; methodology, C.-M.C. and C.-S.L.; software, W.-C.C.; validation, C.-S.L., W.-C.C. and Y.-L.H.; formal analysis, C.-S.L. and W.-C.C.; investigation, C.-M.C.; resources, C.-S.L.; data curation, C.-T.C.; writing—original draft preparation, C.-S.L.; writing—review and editing, C.-S.L. and C.-M.C.; visualization, C.-M.C.; supervision, C.-S.L.; project administration, C.-S.L., Y.-L.H., and C.-M.C.; funding acquisition, C.-S.L., C.-M.C., and Y.-L.H. All authors have read and agreed to the published version of the manuscript.

**Funding:** This research project was supported by the Ministry of Science and Technology, under Grant No. MOST 108-2628-E-035-002-MY3 and MOST 108-2221-E-035-069.

**Conflicts of Interest:** The authors declare no conflict of interest. The funders had no role in the design of the study; in the collection, analyses, or interpretation of data; in the writing of the manuscript, or in the decision to publish the results.

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
