# Peer review of "Development and Application of a Human–Machine Interface Using Head Control and Flexible Numeric Tables for the Severely Disabled"

_applsci, doi:10.3390/app10197005_

Round 1
Reviewer 1 Report
This is an interesting technical paper that demonstrates how head movements can be detected to make inputs to a computer that can be useful for people with disabilities.
The paper generally reads well, and the general concept can be appreciated.
In section 2, it could be stated what hardware and software is needed to run the system e.g. a webcam, image recognition software, etc.
I struggled with understanding the equations used to calculate the head direction, on pages 2 and 4. To assist with this, I would suggest the following changes:
(1) Fig 2 shows the movement of the head on an x,y chart but it would be useful if the outline of the head could be shown on this as an outline in the two positions on this chart.
(2) Consider writing the mathematical content for (4), (5) and (6) in words rather than as equations to make it easier for reader. Perhaps the equations themselves can be put into an appendix.
In Fig 5, I can understand that the head might move in eight directions so was unsure what the difference between the two parts of each of the 8 directions meant i.e. 51 and 52, 31 and 32. It would also be useful to include an English version of Fig. 5 or to state in the text what some of the sentences mean.
It should be stated how many people took part in the experiment to test the algorithm.
As well as tracking head movements, does the system need an activation command so the user moves their head to select a sentence, do they need to select ‘OK’ in some way?
Typos: p2 ‘who only managed’, ‘Lalithamani used a single’, ‘the system proceeds’.
Author Response
- In section 2, it could be stated what hardware and software is needed to run the system e.g. a webcam, image recognition software, etc.
Response: In Section 2, Page. 2, Line. 34, the explanation was added as follows:
This research uses a personal computer and C++ object-oriented development platform for program development. Under Microsoft's Windows operating system, the user's control speed is improved with simple operation methods.
I struggled with understanding the equations used to calculate the head direction, on pages 2 and 4. To assist with this, I would suggest the following changes:
(1) Fig 2 shows the movement of the head on an x,y chart but it would be useful if the outline of the head could be shown on this as an outline in the two positions on this chart.
Response: Figure 2 had been modified.
(2) Consider writing the mathematical content for (4), (5) and (6) in words rather than as equations to make it easier for reader. Perhaps the equations themselves can be put into an appendix.
Response: The equations had been modified. |
It can be extended to judge the right upward, right downward, left upward, and left downward movements of the head (Fig. 4). Where w, v are the Adjustment parameters, Where w, v are the Adjustment parameters. Usually set w =20 and v=10 for initialization.
|
|
In Fig 5, I can understand that the head might move in eight directions so was unsure what the difference between the two parts of each of the 8 directions meant i.e. 51 and 52, 31 and 32. It would also be useful to include an English version of Fig. 5 or to state in the text what some of the sentences mean. It should be stated how many people took part in the experiment to test the algorithm.
Response: In Page. 6, Line. 17, the explanation was added as follows:
“It can obtain the eight postures of up, down, left, right, top right, bottom right, top left, and bottom left, then converts the result of the feature point position detected by the system to judge. For example, the English letter "E" has a code of two-numeric-code 15. The user needs to complete the first head action selection 1 and then the second head action selection 5 to complete the input of the English letter "E".”
As well as tracking head movements, does the system need an activation command so the user moves their head to select a sentence, do they need to select ‘OK’ in some way?
Response: In Page. 6, Line. 22, the explanation was added as follows:
“As well as tracking head movements, the system needn’t an extra activation command so the user moves their head to select a sentence. When a command is made by mistake, the user only needs to shake his head and do it again,”

Reviewer 2 Report
The article presents the concept of an interface for entering information into a computer for disabled people communicating through head movements. A method of detecting one of eight types of movement was proposed: down, up, left, right, and combinations of these shifts. The detected movement or a sequence of several movements is translated into appropriate characters or words according to a predetermined table. Head position detection is based on the detection of skin-colored pixels (by assuming a specific, constant range of R, G, B components in the RGB color space), determining the center of the area of ​​these pixels and making decisions based on the change in the number of pixels and the shift of the computed center of the area.
In my opinion, such assumptions greatly limit the possibility of using this method in non-laboratory conditions, i.e. other than a carefully prepared environment, background, properly dressed person sitting in a precise position and distance to the camera. The motion detection criteria given in the article contain specific pixel coordinates, where we expect an image of a tilted head (formulas 5, 6). Obtaining such positioning repeatability under practical conditions and with the use of other equipment, e.g. a camera with a different focal length, a person sitting closer or further from the camera, other lighting, etc. is difficult.
I propose to consider using another method of face position detection, eg Haar-like features for face detection. Such methods have been used for many years, for example in cameras to find a face in the scene and determine the position to focus on or to decide when the photo should be taken - when the subject smiles. Detection of the face position is fast (it works in real time) and does not require special conditions such as a specific background, the position of the person relative to the lens, etc. After determining the area containing the face, you can also use the motion detection criteria. However, I would avoid giving absolute coordinates in such criteria - these should be values ​​depending on the size of the face image and the current location.
Some remarks:
Line 55: vec- tor
77-78 Formula 1 assumes that in RGB space the skin color is cuboid, which is a simplification.
79 Why "Rxy, Gxy, and Bxy are the gray level of RGB channels" - they are just values ​​of R, G, B components
84 Repeated word: "the system of the system proceeds"
92 Figure 1 shows the results of thresholding - skin-colored area detection. Why does the area marked in blue appear as the result of analyzing an image with a much lower resolution than the original? This looks roughly 32 * 32 pixels resolution, much lower than the resolution of the image
105 Why should the "skin color highlights" number change when the head is moved? It follows from the following conditions that it should decrease when moving up or down and remain the same or increase after moving left or right.
In Figure 3, fewer blue pixels are actually marked for (a) and (b), i.e., up (a) and down (b) movements. But why (in Figure (a)) are there so many pixels belonging to the face unrecognized as skin?
If the area in the color of the skin was marked "by hand", it would be much larger than in the reference drawing (Fig. 1) - the neck is visible. As the head moves downwards (b), the area will actually be smaller.
Further, in condition (6), the same Uc pixel count criterion is used
to distinguish a "shift down" from a "shift up" in the following combinations: left-top, left-bottom, right-top, right-bottom (note: in the second line of formula (6) there should be Uc = 1 instead of Ux = 1). It would be useful to justify this as Figure 4 of these situations does not show significant differences in the size of the blue areas.
Tables 1 and 2 - Are these two-digit or three-digit codes assigned to each character? Does the user make two or three head movements to express this code?
Figures 6, 7, 8 do not describe the axis - it is not clear what is shown
in these figures.
Author Response
In my opinion, such assumptions greatly limit the possibility of using this method in non-laboratory conditions, i.e. other than a carefully prepared environment, background, properly dressed person sitting in a precise position and distance to the camera. The motion detection criteria given in the article contain specific pixel coordinates, where we expect an image of a tilted head (formulas 5, 6). Obtaining such positioning repeatability under practical conditions and with the use of other equipment, e.g. a camera with a different focal length, a person sitting closer or further from the camera, other lighting, etc. is difficult.
I propose to consider using another method of face position detection, eg Haar-like features for face detection. Such methods have been used for many years, for example in cameras to find a face in the scene and determine the position to focus on or to decide when the photo should be taken - when the subject smiles. Detection of the face position is fast (it works in real time) and does not require special conditions such as a specific background, the position of the person relative to the lens, etc. After determining the area containing the face, you can also use the motion detection criteria. However, I would avoid giving absolute coordinates in such criteria - these should be values ​​depending on the size of the face image and the current location.
Response: In Section 2, Page. 2, Line. 34, the explanation was added as follows: This research mainly uses image sensing equipment to detect head rotation position information to complete the comparison of head posture changes. Due to the use of a telescope lens, the user does not need to be very close to the camera to get a clear head image, which can save processing time for face or head detection. A personal computer and C++ object-oriented development platform are used for program development. Under Microsoft's Windows operating system, the user's control speed is improved with simple operation methods. In the aspect of searching for the target object (the head), first we adjust the proper skin color threshold value for the image input and at the same time calculate the coordinate of the center of the head (Fig. 1). Figure 2 had been modified so the outline of the head could be shown on this as an outline in the two positions on this chart. The equations had been modified. |
Some remarks:
Line 55: vec- tor
Response: The error had been modified.
77-78 Formula 1 assumes that in RGB space the skin color is cuboid, which is a simplification.
Response: In Line. 89, the explanation was added as follows:
Equation (1) assumes that in RGB space the skin color is cuboid, which is a simplification.
79 Why "Rxy, Gxy, and Bxy are the gray level of RGB channels" - they are just values ​​of R, G, B components.
In Line. 87, the explanation was modified as follows:
“where Rxy, Gxy, and Bxy are the value of RGB components.”
84 Repeated word: "the system of the system proceeds"
Response: The error had been modified.
92 Figure 1 shows the results of thresholding - skin-colored area detection. Why does the area marked in blue appear as the result of analyzing an image with a much lower resolution than the original? This looks roughly 32 * 32 pixels resolution, much lower than the resolution of the image
Response: Figure 1 had been modified.
105 Why should the "skin color highlights" number change when the head is moved? It follows from the following conditions that it should decrease when moving up or down and remain the same or increase after moving left or right.
In Figure 3, fewer blue pixels are actually marked for (a) and (b), i.e., up (a) and down (b) movements. But why (in Figure (a)) are there so many pixels belonging to the face unrecognized as skin?
Response: In Line. 121, the explanation was modified as follows:
Here, (Ux, Uy) are respectively the component x and component y of the datum point; (Ux2, Uy2) are the respective component x and component y of the barycenter coordinate of the image after its movement. When the head is raised or lowered, due to light reflection, many pixels belonging to the face unrecognized as skin, so the system must have a fault-tolerant design. After the above judgment of head direction, the system will be able to judge the upward, downward, leftward, and rightward movements of the head. In Fig. 3, the blue parts show the facial positions traced by the system. The background is normally the wall of an office room. It can be defined as the follows.
If the area in the color of the skin was marked "by hand", it would be much larger than in the reference drawing (Fig. 1) - the neck is visible. As the head moves downwards (b), the area will actually be smaller.
Response: In Line. 130, the explanation was modified as follows:
The area in the color of the skin would be much larger in the case of the neck is visible. As the head moves downwards, the area will actually be smaller. The adjustment parameters can be applied here to solve this problem.
Further, in condition (6), the same Uc pixel count criterion is used
to distinguish a "shift down" from a "shift up" in the following combinations: left-top, left-bottom, right-top, right-bottom (note: in the second line of formula (6) there should be Uc = 1 instead of Ux = 1). It would be useful to justify this as Figure 4 of these situations does not show significant differences in the size of the blue areas.
Response: In Line. 129, the explanation was added as follows:
Where w, v are the Adjustment parameters. w, v are the Adjustment parameters. It can be extended to judge the right upward, right downward, left upward, and left downward movements of the head (Fig. 4). Usually set w =20 and v=10 for initialization. The area in the color of the skin would be much larger in the case of the neck is visible. As the head moves downwards, the area will actually be smaller. The adjustment parameters can be applied here to solve this problem. Although these situations of Fig. 4 do not show significant differences in the size of the blue areas. But through the innovative judgment method of this system, the head posture in eight directions can still be correctly judged.
Tables 1 and 2 - Are these two-digit or three-digit codes assigned to each character? Does the user make two or three head movements to express this code?
Response: In Page. 6, Line. 17, the explanation was added as follows:
“It can obtain the eight postures of up, down, left, right, top right, bottom right, top left, and bottom left, then converts the result of the feature point position detected by the system to judge. For example, the English letter "E" has a code of two-numeric-code 15. The user needs to complete the first head action selection 1 and then the second head action selection 5 to complete the input of the English letter "E".”
Figures 6, 7, 8 do not describe the axis - it is not clear what is shown in these figures.
Response: Figure6, 7, 8 had been modified.
Round 2
Reviewer 2 Report
Thank you for replying to my comments. Most of them were taken into account. In your response, you wrote that Figure 1 had been changed. But one can still see a lower resolution of the area marked in blue in this figure (and others) than the resolution of the scene image itself. I understand that changing the approach to the problem (a different way of detecting face positions) would be too laborious - I encourage you to experiment with other methods in your next research. The text still requires editorial corrections (alignments in formulas etc.).
Author Response
Revision List (Manuscript ID applsci-947893)
Reviewers' comments:
Thank you for replying to my comments. Most of them were taken into account. In your response, you wrote that Figure 1 had been changed. But one can still see a lower resolution of the area marked in blue in this figure (and others) than the resolution of the scene image itself. I understand that changing the approach to the problem (a different way of detecting face positions) would be too laborious - I encourage you to experiment with other methods in your next research. The text still requires editorial corrections (alignments in formulas etc.).
Response: I am very appreciated for your useful comments. The changes of the methodology, results and discussion of our manuscript are listed as follows.
- The resolution of Figure 1 had been modified.
- The text had been corrected.